# Fast and Explicit: Slice-to-Volume Reconstruction via 3D Gaussian Primitives with Analytic Point Spread Function Modeling

**Maik Dannecker**[*1]                                    M.DANNECKER@TUM.DE
**Steven Jia**[*2]                                          STEVEN.JIA@UNIV-AMU.FR
**Nil Stolt-Ansó**[1]                                        NIL.STOLT@TUM.DE
**Nadine Girard**[2]                                       NADINE.GIRARD@AP-HM.FR
**Guillaume Auzias**[2]                              GUILLAUME.AUZIAS@UNIV-AMU.FR
**François Rousseau**[3]                         FRANCOIS.ROUSSEAU@IMT-ATLANTIQUE.FR
**Daniel Rueckert**[1,4]                                 DANIEL.RUECKERT@TUM.DE

[1] *TUM University Hospital, Technical University of Munich, Munich, Germany*

[2] *Institut de Neurosciences de la Timone, Aix-Marseille Université, Marseille, France*

[3] *IMT Atlantique, Brest, France*

[4] *Department of Computing, Imperial College London, London, UK*

**Editors:** Accepted for publication at MIDL 2026

## Abstract

Recovering high-fidelity 3D images from sparse or degraded 2D images is a fundamental challenge in medical imaging, with broad applications ranging from 3D ultrasound reconstruction to MRI super-resolution. In the context of fetal MRI, high-resolution 3D reconstruction of the brain from motion-corrupted low-resolution 2D acquisitions is a prerequisite for accurate neurodevelopmental diagnosis. While implicit neural representations (INRs) have recently established state-of-the-art performance in self-supervised slice-to-volume reconstruction (SVR), they suffer from a critical computational bottleneck: accurately modeling the image acquisition physics requires expensive stochastic Monte Carlo sampling to approximate the point spread function (PSF). In this work, we propose a shift from neural network based implicit representations to Gaussian based explicit representations. By parameterizing the HR 3D image volume as a field of anisotropic Gaussian primitives, we leverage the property of Gaussians being closed under convolution and thus derive a *closed-form analytical solution* for the forward model. This formulation reduces the previously intractable acquisition integral to an exact covariance addition ($\Sigma_{obs} = \Sigma_{HR} + \Sigma_{PSF}$), effectively bypassing the need for compute-intensive stochastic sampling while ensuring exact gradient propagation. We demonstrate that our approach matches the reconstruction quality of self-supervised state-of-the-art SVR frameworks while delivering a $5\times$–$10\times$ speed-up on neonatal and fetal data. With convergence often reached in under 30 seconds, our framework paves the way towards translation into clinical routine of real-time fetal 3D MRI. Code will be public at https://github.com/m-dannecker/Gaussian-Primitives-for-Fast-SVR.

**Keywords:** Slice-to-Volume Reconstruction, 3D Gaussian Splatting, Super-Resolution, Fetal MRI, Neonatal MRI.

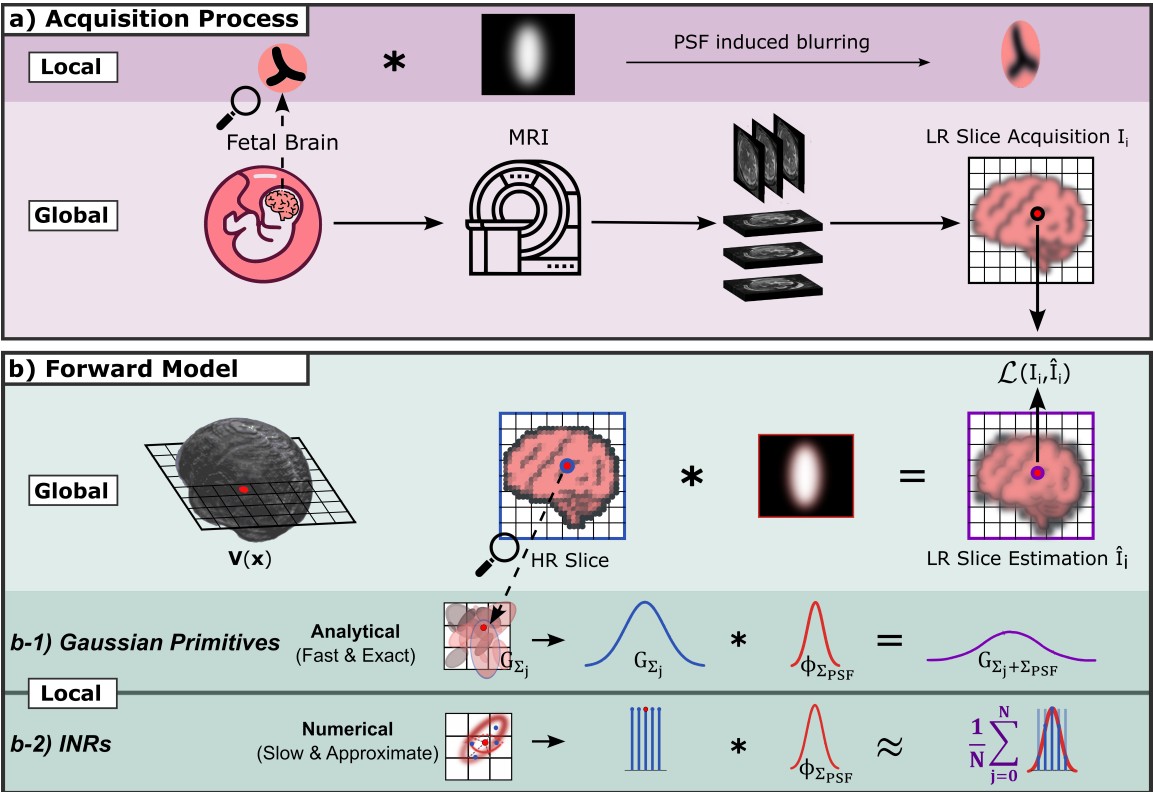

Figure 1: **SVR via Gaussian Primitives. a)** Clinical acquisition introduces through-slice blurring due to the anisotropic thick-slice PSF. **b)** We parameterize the unknown HR volume as a field of Gaussians $G_{\Sigma_j}$. By approximating the PSF as a Gaussian kernel $\phi_{\Sigma_{PSF}}$, we exploit the closure property that the convolution of two Gaussians is strictly another Gaussian. Consequently, the observed low-resolution slice $\hat{I}_i$ is simulated analytically via exact covariance addition ($\Sigma_{obs} = \Sigma_j + \Sigma_{PSF}$) (**b-1**), replacing expensive Monte Carlo sampling (**b-2**) with a fast, deterministic forward pass. Note, motion correction is omitted in this figure for clarity.

## 1. Introduction

The reconstruction of continuous 3D anatomical models from sparse or low-dimensional measurements is a ubiquitous challenge across medical imaging, essential for tasks ranging from 3D ultrasound rendering to image super-resolution. In the specific domain of perinatal neuroradiology, the reconstruction of the moving fetal and neonatal brain from stacks of thick, low-resolution (LR) 2D MRI slices is increasingly critical for the early diagnosis of neurodevelopmental disorders (Uus et al., 2023). Consequently, slice-to-volume reconstruction (SVR) is required to recover a coherent, isotropic high-resolution volume from these sparse, motion-corrupted stacks. Speed is a crucial factor in this pipeline; rapid reconstruc-

---

* Contributed equally

tion allows clinicians to assess scan quality in real-time and re-acquire corrupted stacks during the same examination, avoiding the logistical burden of patient recall.

Classic SVR frameworks (Jiang et al., 2007; Tourbier et al., 2015; Ebner et al., 2020) formulate reconstruction as a discrete optimization problem on a fixed voxel grid. While effective, these methods are limited by discrete sampling resolutions and rely on scattered data interpolation. Recently, coordinate-based networks or implicit neural representations (INRs), such as NeSVoR (Xu et al., 2023), have advanced the state-of-the-art by modeling the 3D image volume as a continuous function parameterized by a Multilayer Perceptron (MLP)(Wu et al., 2021; McGinnis et al., 2023).

Recovering a 3D volume from 2D views is a fundamental task in computer graphics, driving the rapid translation of Neural Radiance Fields (Mildenhall et al., 2022) to MRI. Notably, NeSVoR successfully adapted the hash encodings of Instant-NGP (Müller et al., 2022) to accelerate fetal brain SVR. Recently, however, *3D Gaussian Splatting (3DGS)* (Kerbl et al., 2023) has surpassed implicit ray-marching methods, demonstrating superior training efficiency and rendering quality. Motivated by these advances, we explore the translation of this explicit Gaussian representation to medical reconstruction.

Although INRs offer resolution independence, they suffer from a severe computational bottleneck in the forward model. To accurately simulate the physics of slice acquisition with anisotropic resolution (high in-slice but low out-of-slice resolution), the network must integrate the continuous volumetric signal over the slice profile, typically approximated by a Gaussian point spread function (PSF) (Rousseau et al., 2005). For implicit networks, this convolution is analytically intractable and requires Monte Carlo integration (stratified sampling) with up to 256 samples per query point (Xu et al., 2023). As each sample constitutes a forward pass, this stochastic approximation causes considerable compute overhead in current SVR frameworks.

### 1.1. Contribution

We reformulate SVR from an implicit representation to an explicit Gaussian representation for fetal MRI, as illustrated in Figure 1. Drawing inspiration from 3DGS and Image-GS (Zhang et al., 2025), we formulate the HR 3D brain image as a sparse cloud of anisotropic Gaussians. As our task is volumetric (3D $\rightarrow$ 3D) rather than projective (3D $\rightarrow$ 2D), we eliminate the need for rasterization and projection matrices; we refer to this representation as *Gaussian Primitives*. This formulation replaces the intractable slice acquisition integral for PSF modeling with an exact, closed-form evaluation, thereby unlocking unprecedented reconstruction speeds. Our specific contributions are:

- **Gaussian Primitives.** We introduce an explicit 3D representation for super-resolution in medical imaging. This continuous, resolution-independent representation enables rapid optimization without the geometric overhead of graphical projection matrices.

- **Analytic PSF.** Contrary to INRs which approximate the PSF through expensive stochastic Monte Carlo sampling, we derive a *closed-form analytical solution* utilizing the closure of Gaussians under convolution.

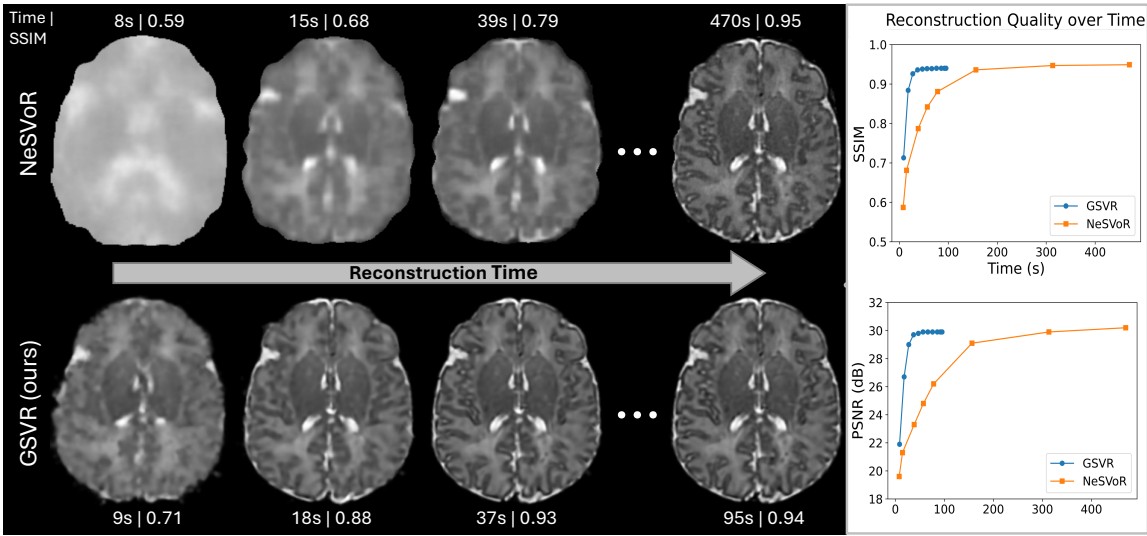

Figure 2: **Convergence Speed.** Comparison of SSIM and PSNR over time (seconds) of the proposed GSVR vs. NeSVoR performed on one simulated subjected.

- **Scale Regularization.** We introduce Gaussian scale regularization to enforce a piecewise smooth prior, mitigating overfitting to noise and preventing high-frequency artifacts.

- **Efficiency.** We demonstrate reconstruction speed-ups of factor $5 - 10$ compared to INR baselines on neonatal data, with convergence often reached in under 30 seconds, facilitating preview reconstructions during scan-time (Figure 2).

## 2. Methodology

We formulate the SVR problem as the inverse recovery of a canonical, HR, isotropic volume $V : \mathbb{R}^3 \to \mathbb{R}$ from a collection of motion-corrupted, anisotropic, LR slices $\mathcal{I} = \{I_i\}_{i=1}^M$, as depicted in Figure 1.

### 2.1. Physical Forward Model

We adopt the continuous acquisition model of NeSVoR. In this model, an observed intensity $I_i(\mathbf{u})$ at a 2D pixel coordinate $\mathbf{u} \in \mathbb{R}^2$ on the $i$-th slice is the result of integrating the underlying image volume $V$ over the slice profile (PSF), subject to motion:

$$I_i(\mathbf{u}) = \sigma_i \left( \int_{\mathbb{R}^3} V(\mathbf{x}) \cdot \phi_i(\mathbf{x} - \mathcal{T}_i(\mathbf{u})) \, d\mathbf{x} \right) + \epsilon_i \tag{1}$$

Here $\mathcal{T}_i(\mathbf{u}) = \mathbf{R}_i \mathbf{u} + \mathbf{t}_i$ maps the 2D slice coordinate to 3D world space via a rigid transformation (rotation $\mathbf{R}_i$, translation $\mathbf{t}_i$), $\sigma_i$ represents the slice-specific intensity scaling, $\epsilon_i$ represents Gaussian noise, and $\phi_i$ denotes the 3D PSF oriented according to the slice geometry.

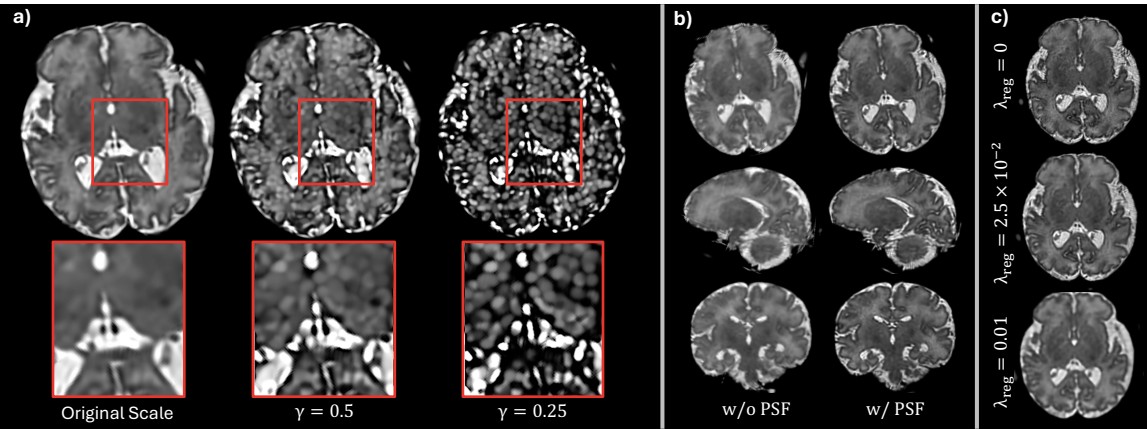

Figure 3: **a) Gaussian Representation:** Shrinking the Gaussians by factor $\gamma$ during inference reveals content adaptation: isotropic primitives cover homogeneous regions, while anisotropic ellipsoids capture boundaries. **b) Analytic PSF:** Explicitly modeling the PSF resolves slice thickness, whereas its absence results in blur and partial volume artifacts. **c) Regularization:** Increasing $\lambda_{reg}$ prevents noise overfitting by penalizing very small Gaussians. Large $\lambda_{reg}$ results in over-smoothing.

For INRs, Equation (1) has no closed-form solution. Existing methods rely on stochastic Monte Carlo sampling to approximate the PSF, typically requiring $> 100$ queries/samples per point, which is the primary limiting factor for reconstruction speed (Xu et al., 2023).

## 2.2. Explicit Gaussian Representation

We propose an explicit Gaussian representation to circumvent the sampling bottleneck inherent to INRs. This shift enables a closed-form analytical solution to the acquisition integral of Equation (1), which we detail in Section 2.3.

In contrast to standard 3D Gaussian Splatting (3DGS) (Kerbl et al., 2023) that relies on 2D rasterization, we model the MRI signal as continuous volumetric field of Gaussian primitives. Formally, we define the volume $V(\mathbf{x})$ as a mixture of $N$ 3D Gaussian primitives (see Figure 3-a). Each primitive $G_j$ is defined by 11 learnable parameters: mean position $\boldsymbol{\mu}_j \in \mathbb{R}^3$, a covariance matrix $\boldsymbol{\Sigma}_j \in \mathbb{R}^{3 \times 3}$ factorized as $\Sigma = RSS^T R^T$ where $S$ is a diagonal scaling matrix and $R$ is a rotation matrix expressed with 4 quaternions, and an intensity $c_j$. The evaluation of the signal at any 3D coordinate $\mathbf{x}$ is computed as a normalized weighted sum:

$$V(\mathbf{x}) = \frac{\sum_{j \in \mathcal{S}^K(\mathbf{x})} c_j \cdot \exp\left(-\frac{1}{2}(\mathbf{x} - \boldsymbol{\mu}_j)^T \boldsymbol{\Sigma}_j^{-1}(\mathbf{x} - \boldsymbol{\mu}_j)\right)}{\sum_{j \in \mathcal{S}^K(\mathbf{x})} \exp\left(-\frac{1}{2}(\mathbf{x} - \boldsymbol{\mu}_j)^T \boldsymbol{\Sigma}_j^{-1}(\mathbf{x} - \boldsymbol{\mu}_j)\right) + \delta} \tag{2}$$

where $\delta$ is a small constant for numerical stability, and $\mathcal{S}^K(\mathbf{x})$ denotes the set of $K$ Gaussians nearest to $\mathbf{x}$ (see Section 2.5).

Our formulation differs from standard 3DGS by strictly operating in 3D world space. Unlike graphical rendering, we do not project primitives onto a 2D imaging plane, eliminating the need for the Jacobian projection matrix. Instead of "splatting" or "rasterizing" Gaussians, we evaluate the continuous volumetric density field directly.

### 2.3. Closed-Form PSF Modeling via Convolution

To accurately resolve the acquisition physics, we must model the PSF, $\phi$, which incorporates the slice selection profile and in-plane blurring. Conceptually, the estimated intensity $\hat{I}_i(\mathbf{u})$ is the result of convolving the underlying volume $V$ at motion-corrected position $\mathcal{T}_i(\mathbf{u})$, with the slice-oriented PSF $\phi_i$:

$$\hat{I}_i(\mathbf{u}) = (V * \phi_i)(\mathcal{T}_i(\mathbf{u})). \tag{3}$$

In standard approaches, including INRs, evaluating this convolution is analytically intractable and typically approximated via compute-expensive Monte Carlo integration (Figure 1, b-2). We circumvent this bottleneck by leveraging the *semi-group property* of Gaussians: *the convolution of two Gaussians is strictly another Gaussian* (Figure 1, b-1).

First, we model the physical PSF as anisotropic 3D Gaussian kernel $\phi = \mathcal{N}(\mathbf{0}, \boldsymbol{\Sigma}_{\text{PSF}})$, where $\boldsymbol{\Sigma}_{\text{PSF}}$ is defined in the slice coordinate system to account for slice resolution and thickness (Rousseau et al., 2010; Gholipour et al., 2010). Second, we parametrize the unknown HR image volume $V$ as a sum of Gaussians $G_j(\mu_j, \boldsymbol{\Sigma}_j)$, as described by Equation (2). The convolution in Equation (3) now reduces to a closed-form addition of their covariance matrices. For a slice rotated by $\mathbf{R}_i$, the effective *observed* covariance $\boldsymbol{\Sigma}_{obs,j}$ for each Gaussian primitive is given by:

$$\boldsymbol{\Sigma}_{obs,j} = \boldsymbol{\Sigma}_j + \mathbf{R}_i \boldsymbol{\Sigma}_{\text{PSF}} \mathbf{R}_i^T. \tag{4}$$

Substituting Equation (4) into the acquisition model transforms the integral from Equation (1) into a direct, exact evaluation. The final estimated LR intensity $\hat{I}_i(\mathbf{u})$ is computed by replacing the HR covariance $\boldsymbol{\Sigma}_j$ with the convolved covariance $\boldsymbol{\Sigma}_{obs,j}$ in the rendering equation:

$$\hat{I}_i(\mathbf{u}) = \sigma_i \frac{\sum_{j \in \mathcal{S}^K} c_j \exp\left(-\frac{1}{2} \mathbf{v}_j^T \boldsymbol{\Sigma}_{obs,j}^{-1} \mathbf{v}_j\right)}{\sum_{j \in \mathcal{S}^K} \exp\left(-\frac{1}{2} \mathbf{v}_j^T \boldsymbol{\Sigma}_{obs,j}^{-1} \mathbf{v}_j\right) + \delta}, \tag{5}$$

Here $\mathbf{v}_j = \mathcal{T}_i(\mathbf{u}) - \boldsymbol{\mu}_j$ is the motion corrected coordinate vector relative to the $j$-th Gaussian mean. This formulation evaluates the physically "blurred" signal via a simple matrix addition, ensuring *fast and exact* gradient propagation without stochastic sampling.

### 2.4. Motion Correction and Outlier Handling

We follow the robust optimization strategy of NeSVoR by jointly optimizing the Gaussian parameters and slice-wise rigid transformations. For each slice $i$, we learn a rigid transformation $\mathcal{T}_i(\mathbf{u}) = \mathbf{R}_i \mathbf{u} + \mathbf{t}_i$. Critically, the anisotropic PSF covariance $\boldsymbol{\Sigma}_{\text{PSF}}$ is dynamically rotated alongside the slice via Equation (4), ensuring the through-plane blur remains aligned with the slice normal in 3D space.

To robustly handle outliers (e.g., signal drop, inconsistent contrast), we incorporate two mechanisms into the forward model: a learnable intensity scalar $\sigma_i$ to compensate for global intensity shifts, and an aleatoric uncertainty weight $\omega_i$ to down-weight corrupted slices during loss computation. Together, this setup allows the framework to self-correct for motion and artifacts in an end-to-end manner.

## 2.5. Optimization Strategy

**Sparse Computation (Top-K):** To ensure computational efficiency, we employ a Top-K culling strategy inspired by (Zhang et al., 2025). For any motion corrected coordinate $\mathcal{T}_i(\mathbf{u})$, we only compute contributions of the $K$ nearest Gaussians (by L2 distance $\|\mathcal{T}_i(\mathbf{u}) - \boldsymbol{\mu}_j\|_2$).

**Content-Adaptive Initialization:** To accelerate convergence, we utilize gradient adaptive initialization (Zhang et al., 2025). We compute the gradient magnitude $\|\nabla I_i\|$ of the input slices to construct a sampling probability map, and define the initialization probability $\mathbb{P}_{init}$ as a mixture of gradient-based and uniform sampling:

$$\mathbb{P}_{init}(\mathcal{T}_i(\mathbf{u})) \propto (1 - \lambda_{init})\|\nabla I_i(\mathbf{u})\| + \lambda_{init}. \tag{6}$$

We initialize the means $\boldsymbol{\mu}_j$ by sampling from $\mathbb{P}_{init}$, ensuring high Gaussian density in detailed anatomical regions (tissue boundaries) and lower density in homogeneous regions.

**Loss Function:** We optimize the Gaussian parameters $(\boldsymbol{\mu}, \boldsymbol{\Sigma}, c)$ and motion parameters $(\mathbf{q}, \mathbf{t})$ to minimize the $L_1$ reconstruction loss combined with a scale regularization term

$$\mathcal{L} = \sum_i \|I_i - \hat{I}_i\|_1 + \lambda_{reg} \sum_{j=1}^{N} \|\mathbf{s}_j - \mathbf{s}_{target}\|_2^2, \tag{7}$$

where $I_i$ is the *observed* LR intensity of slice $i$ and $\hat{I}_i$ denotes the intensity predicted by the model according to (5). Finally, $\mathbf{s}_j$ denotes the scaling factors of the covariance and $\mathbf{s}_{target}$ a hyperparameter defining the desired mean scale (empirically set to 1.6mm isotropic). The regularization prevents the Gaussians from becoming too small (overfitting to noise) or expanding excessively, thereby ensuring smooth anatomical continuity.

## 3. Experiments and Results

### 3.1. Baselines

We compare our framework against two *self-supervised* state-of-the-art SVR methods:
**1) SVRTK** (Kuklisova-Murgasova et al., 2012a): The standard CPU-based iterative optimization toolkit, executed here on 16 parallel cores. We limited reconstruction to three iterations, as convergence was observed at this stage. We noted that SVRTK is highly sensitive to masking imperfections; despite manual segmentation corrections, final reconstructions occasionally exhibited field-of-view cut-offs.
**2) NeSVoR** (Xu et al., 2023): A GPU-accelerated INR approach yielding state-of-the-art reconstruction quality and runtime. We utilized default hyperparameters, which proved optimal for our experiments. Notably, NeSVoR optimizes $\approx 4.8$ million parameters, roughly an order of magnitude more than our $\approx 550,000$ parameters.

Table 1: Quantitative comparison of reconstruction accuracy and runtime on 10 neonatal subjects with simulated motion corruption. Best metrics in **bold**.

| Method | PSNR ↑ | SSIM ↑ | NCC ↑ | NRMSE ↓ | Time (s)↓ |
|---|---|---|---|---|---|
| SVRTK (Kuklisova-Murgasova et al., 2012b) | $18.65^* \pm 1.22$ | $0.641^* \pm 0.074$ | $0.541^* \pm 0.126$ | $0.118^* \pm 0.017$ | 573 |
| NeSVoR (Xu et al., 2023) | $28.38 \pm 1.37$ | $0.933 \pm 0.030$ | $0.955 \pm 0.011$ | $0.039 \pm 0.006$ | 478 |
| GSVR (Ours) | $\mathbf{28.76 \pm 1.52}$ | $\mathbf{0.936 \pm 0.018}$ | $\mathbf{0.957 \pm 0.012}$ | $\mathbf{0.037 \pm 0.007}$ | **79** |

*Statistically significant difference to the best performing method ($p < 0.05$, paired t-test).

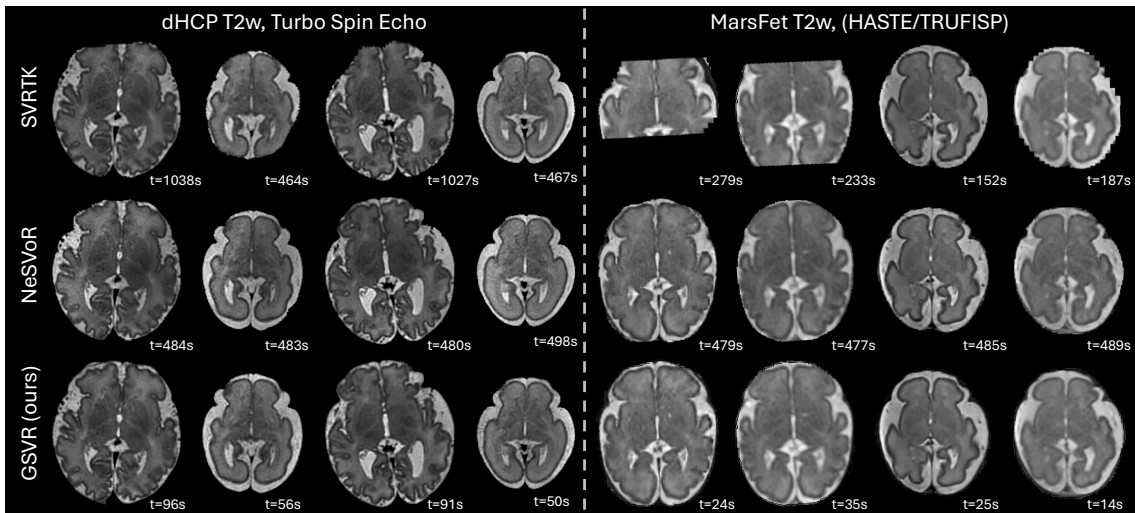

Figure 4: **Qualitative comparison on dHCP and MarsFet cohorts.** SVRTK exhibits masking artifacts on limited FOVs. The proposed GSVR matches the high reconstruction quality of NeSVoR while achieving speed-ups of up to factor 10.

### 3.2. Data

**Simulated Data:** To ensure high-quality ground truth with minimal motion artifacts, we selected isotropic (0.5 mm) neonatal scans from the dHCP dataset (Cordero-Grande et al., 2019). We simulated clinical fetal acquisitions by generating three orthogonal stacks ($0.5 \times 0.5 \times 3$ mm) (Mercier et al., 2025). To model the worst-case residual motion that is typically expected after SVoRT pre-alignment, we applied random slice-wise rigid transformations sampled from $\mathcal{U}(-6°, 6°)$ and $\mathcal{U}(-4, 4)$ mm.

**In-vivo dHCP Fetal Data:** We selected four subjects from the fetal dHCP dataset (Price et al., 2019) between 27 and 37 weeks gestational age (GA). Each subject includes 6 uniquely oriented motion-corrupted stacks centered on the fetal brain using a 3T Philips Achieva with zoomed multiband single-shot Turbo Spin Echo sequence at an in-slice resolution of $1.1 \times 1.1 \text{mm}^2$ and 2.2 mm slice thickness.

**In-vivo MarsFet Fetal Data:** We used two subjects from the clinical dataset named MarsFet(Mihailov et al., 2025). Both subjects include two acquisition sequences, a half-Fourier single-shot turbo spin-echo (HASTE) sequence ($0.74 \times 0.74 \times 3.5$mm) and a TRU-FISP sequence ($0.625 \times 0.625 \times 3.0$mm) (Girard et al., 2003).

**Pre-processing.** We applied a unified pipeline across all methods. In-vivo scans underwent bias field correction (Tustison et al., 2010) (plus denoising for MarsFet), automated brain masking (Ranzini et al., 2021), and were co-registered to a common space via SVoRT (Xu et al., 2022). For simulated data, we used ground-truth masks with no further processing. Target resolution for all data and methods was set to $0.5mm$ isotropic.

## 3.3. Results

**Quantitative Evaluation on Simulated Data:** Table 3 summarizes performance on 10 simulated subjects. Our Gaussian-based SVR (GSVR) matches the state-of-the-art NeSVoR across all metrics (PSNR 28.07, SSIM 0.936) without external slice pre-alignment, demonstrating robust internal motion correction. SVRTK degrades significantly (18.20 dB) due to sensitivity to large motion corruption (see Appendix Figure 5). Critically, GSVR completes reconstruction in just 79s, a $6\times$ and $7\times$ speed-up over NeSVoR (478s) and SVRTK (573s), respectively. As shown in Figure 2, our Gaussian representation with analytic PSF enables rapid convergence, yielding usable previews (SSIM $> 0.8$) in under 20 seconds.

**Qualitative Evaluation on In-vivo Data:** Figure 4 shows reconstructions on real-world clinical scans from the dHCP and MarsFet datasets. SVRTK reconstructions often exhibit field-of-view (FOV) cut-offs, failing to recover the complete brain due to masking failures or incomplete FOV in the raw acquisitions. Both NeSVoR and GSVR successfully recover coherent, high-resolution isotropic 3D volumes. Our method resolves fine anatomical structures, such as the cortex, with a level of sharpness comparable to NeSVoR. The visual results further confirm the efficiency gains of GSVR demonstrating speed-ups of up to factor 10 over NeSVoR an SVRTK without compromising reconstruction quality.

## 3.4. Ablation Study

We analyze the impact of model components in Table 2.

**Gaussian Density ($N$):** Increasing the number of primitives from $10k$ to $50k$ improves detail recovery considerably ($+1.4$ dB); further increases yield diminishing returns.

**Sparsity ($Top$-$K$):** Reducing neighbors to $K = 10$ degrades reconstruction ($-2.7$ dB) by spatially limiting the gradient flow required for robust motion correction. $K = 80$ yields the highest metrics, but increases runtime by 40%; we select $K = 50$ as the best trade-off.

**PSF & Regularization:** Disabling the analytic PSF causes a massive drop in PSNR ($-8.0$ dB) due to partial volume effects. Scale regularization ($+1.3$ dB) and adaptive initialization ($+1.3$ dB) prove beneficial to prevent overfitting and ensure robust convergence.

## 4. Discussion

**Analytic Tractability & Broader Impact:** We demonstrated that the computational bottleneck of MRI physics simulation—specifically PSF convolution—can be solved analytically by adopting an explicit Gaussian representation. Unlike INRs, which require expensive

Table 2: Ablation study analyzing the impact of Gaussian density ($N_{Gauss}$), sparsity (top-$K$), and model components on reconstruction quality and runtime.

| Configuration | | | | | Metrics | | |
|---|---|---|---|---|---|---|---|
| $N_{Gauss}$ | Top-$K$ | $\text{Init}_{adapt}$ | PSF | Reg | PSNR ↑ | SSIM ↑ | Time (s) ↓ |
| 10000 | 50 | ✓ | ✓ | ✓ | $27.41 \pm 1.31$ | $0.906 \pm 0.029$ | 64 |
| 20000 | 50 | ✓ | ✓ | ✓ | $28.38 \pm 1.42$ | $0.927 \pm 0.020$ | 69 |
| 50000 | 50 | ✓ | ✓ | ✓ | $28.76 \pm 1.52$ | $0.936 \pm 0.018$ | 79 |
| 80000 | 50 | ✓ | ✓ | ✓ | $28.60 \pm 1.59$ | $0.936 \pm 0.018$ | 82 |
| 50000 | 10 | ✓ | ✓ | ✓ | $26.07 \pm 1.02$ | $0.900 \pm 0.018$ | **28** |
| 50000 | 20 | ✓ | ✓ | ✓ | $27.57 \pm 1.25$ | $0.923 \pm 0.016$ | 40 |
| 50000 | 80 | ✓ | ✓ | ✓ | $\mathbf{28.88 \pm 1.51}$ | $\mathbf{0.939 \pm 0.018}$ | 111 |
| 50000 | 50 | ✗ | ✓ | ✓ | $27.50 \pm 0.92$ | $0.920 \pm 0.019$ | 80 |
| 50000 | 50 | ✓ | ✗ | ✓ | $20.78 \pm 1.02$ | $0.723 \pm 0.049$ | 69 |
| 50000 | 50 | ✓ | ✓ | ✗ | $27.42 \pm 1.71$ | $0.917 \pm 0.032$ | 78 |

Shaded row indicates model configuration used in this study.

stochastic Monte Carlo sampling to approximate slice integration, our approach transforms this into a closed-form algebraic operation ($\mathbf{\Sigma}_{obs} = \mathbf{\Sigma}_{HR} + \mathbf{\Sigma}_{PSF}$). This finding could have broad implications beyond SVR: conceptually, any inverse problem involving Gaussian kernels—such as deconvolution microscopy, PET partial volume correction, or diffusion tensor estimation—could benefit from this formulation, restoring the exact gradient propagation of classical signal processing within a learning-based framework. However, further exploration in this direction remains future work.

**Towards Real-Time SVR:** By eliminating the sampling bottleneck, our framework achieves usable reconstructions in under 30 seconds, enabling *intra-session quality control*. Clinicians can thus assess scan utility immediately, potentially reducing patient recall rates. While our current implementation relies on generic PyTorch kernels, runtimes reported in 2D Gaussian fitting benchmarks (Zhang et al., 2025) suggest that porting our volumetric operations to dedicated CUDA backends could yield a further 2–4× speed-up. Unlocking this engineering potential paves the way for *real-time SVR*, transforming reconstruction from offline post-processing into an interactive clinical tool.

### 4.1. Limitations and Future Work

**Bias Correction & Robustness:** Future work could integrate bias field correction using low-frequency Gaussian primitives. Additionally, a systematic evaluation is needed to analyze the robustness of the model to severely corrupted acquisitions.

**Coarse-to-Fine Optimization:** Implementing progressive densification of Gaussians primitives (Kerbl et al., 2023)—starting with sparse, large primitives—would enable robust and fast global gradient flow for motion correction even with small neighborhoods, followed by adaptive splitting to reconstruct fine details.

**Extension to Diffusion MRI:** By adapting Spherical Harmonics (used in 3DGS for view-dependence) to model direction-dependent signals, the framework could be extended to reconstruct Diffusion Weighted Imaging data.

## Acknowledgments

This research has been supported by the ERA-NET NEURON MULTI-FACT Project. The project leading to this publication has received funding from the Excellence Initiative of Aix-Marseille Université - A*Midex, a French "Investissements d'Avenir programme" AMX-21-IET-017. Data were provided by the developing Human Connectome Project, KCL-Imperial-Oxford Consortium funded by the European Research Council under the European Union Seventh Framework Programme (FP/2007-2013) / ERC Grant Agreement no. [319456]. We are grateful to the families who generously supported this trial.

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

## Appendix A. Implementation

For initialization, we utilize the content-adaptive strategy described in Sec. 2.5. We set the number of Gaussians to $N = 50,000$ for all experiments. We set the initialization mixing parameter $\lambda_{init} = 0.0$, favoring high-frequency edge regions entirely over uniform distribution to capture anatomical detail rapidly.

### A.1. Optimization Details

The model is trained for 500 epochs using the AdamW optimizer(Loshchilov and Hutter, 2017). We utilize a large effective batch size (processing up to $10^7$ points per pass) to approximate full-batch gradient descent. The learning rates are set as follows: positions $\mu$ and scaling $s$ at $2.5 \times 10^{-2}$, rotation quaternions $q$ and color $c$ at $1.0 \times 10^{-2}$. The motion correction parameters are optimized with a lower learning rate for rotations ($2.5 \times 10^{-3}$) compared to translations ($1.0 \times 10^{-1}$). A StepLR scheduler is applied, decaying the learning rate by a factor of 0.5 every 200 epochs.

We employ the Top-K approximation with $K = 50$ neighbors. To balance speed and accuracy, the neighbor index is updated every 50 epochs using the FAISS library (Johnson et al., 2019) for efficient GPU-based nearest neighbor search. The scale regularization weight was empirically set to $\lambda_{reg} = 2.5 \times 10^{-3}$ with a target scale of $1.6mm$ to encourage smoothness.

### A.2. Acceleration

To eliminate Python overhead in the critical path, we implemented custom fused kernels for:

- **Fused Mahalanobis Distance:** A single kernel computes the covariance inversion and the quadratic form $(\mathbf{x} - \mu)^T \mathbf{\Sigma}^{-1} (\mathbf{x} - \mu)$ for the $K$ nearest neighbors.

- **Fused Motion Correction:** We fuse the quaternion-to-matrix conversion, coordinate transformation, and PSF covariance rotation $\mathbf{R}\mathbf{\Sigma}_{PSF}\mathbf{R}^T$ into a single operation.

This optimized implementation allows for convergence in under 60 seconds for standard fetal volumes.

## Appendix B. Additional Results

Table 3: Quantitative comparison of reconstruction accuracy and runtime on 10 neonatal subjects with simulated motion corruption. Performance under different levels of motion corruption are reported. With no motion corruption, NeSVoR slightly outperforms GSVR which we mainly attribute to the factor 10 higher number of parameters of NeSVoR ($5 \times 10^6$ vs $0.5 \times 10^6$). (Best metrics in **bold**).

| Method | PSNR ↑ | SSIM ↑ | NCC ↑ | NRMSE ↓ | Time (s)↓ |
|---|---|---|---|---|---|
| Normal Motion Corruption (factor 1) | | | | | |
| SVRTK (Kuklisova-Murgasova et al., 2012b) | $18.65^* \pm 1.22$ | $0.641^* \pm 0.074$ | $0.541^* \pm 0.126$ | $0.118^* \pm 0.017$ | 573 |
| NeSVoR (Xu et al., 2023) | $28.38 \pm 1.37$ | $0.933 \pm 0.030$ | $0.955 \pm 0.011$ | $0.039 \pm 0.006$ | 478 |
| GSVR (Ours) | $\mathbf{28.76 \pm 1.52}$ | $\mathbf{0.936 \pm 0.018}$ | $\mathbf{0.957 \pm 0.012}$ | $\mathbf{0.037 \pm 0.007}$ | **79** |
| Small Motion Corruption (factor 1/2) | | | | | |
| SVRTK (Kuklisova-Murgasova et al., 2012b) | $23.00^* \pm 1.06$ | $0.860^* \pm 0.034$ | $0.841^* \pm 0.030$ | $0.071^* \pm 0.009$ | 545 |
| NeSVoR (Xu et al., 2023) | $30.27 \pm 1.61$ | $0.939 \pm 0.038$ | $\mathbf{0.971 \pm 0.011}$ | $0.031 \pm 0.006$ | 469 |
| GSVR (Ours) | $\mathbf{30.43 \pm 1.19}$ | $\mathbf{0.949 \pm 0.020}$ | $\mathbf{0.971 \pm 0.009}$ | $\mathbf{0.030 \pm 0.004}$ | **81** |
| No Motion Corruption (factor 0) | | | | | |
| SVRTK (Kuklisova-Murgasova et al., 2012b) | $25.51^* \pm 0.73$ | $0.922^* \pm 0.011$ | $0.910^* \pm 0.015$ | $0.053^* \pm 0.005$ | 536 |
| NeSVoR (Xu et al., 2023) | $\mathbf{34.61 \pm 0.76}$ | $\mathbf{0.984 \pm 0.001}$ | $\mathbf{0.990 \pm 0.001}$ | $\mathbf{0.019 \pm 0.002}$ | 487 |
| GSVR (Ours) | $33.52^* \pm 1.11$ | $0.980^* \pm 0.003$ | $0.987^* \pm 0.002$ | $0.021^* \pm 0.003$ | **77** |

*Statistically significant difference to the best performing method ($p < 0.05$, paired t-test).

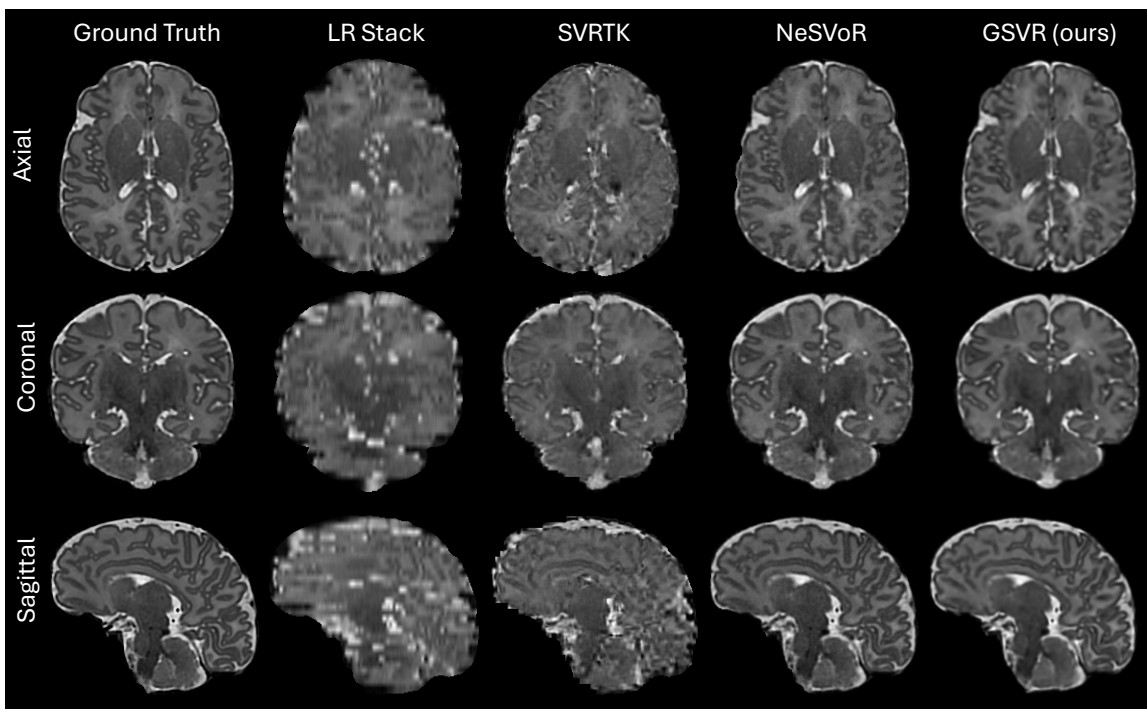

Figure 5: **Qualitative evaluation on simulated data.** Visualizing from left to right the ground truth neonaten reconstruction, the low-resolution (LR) motion corrupted stacks, and the three evaluated methods. Whereas SVRTK struggles with motion correction, NeSVoR and the proposed GSVR framework achieve high quality reconstructions. Reconstruction times for the depicted neonatal subject are 763s (SVRTK), 471s (NeSVoR), and 92s (GSVR).

Table 4: Detailed quantitative comparison of reconstruction accuracy per subject. Time is currently set to 0 as a placeholder. Best metrics per subject are shown in **bold**.

| Subject | Method | PSNR ↑ | SSIM ↑ | NCC ↑ | Time (s)↓ |
|---------|--------|--------|--------|--------|-----------|
| CC00389XX19 | SVRTK | 18.13 | 0.616 | 0.580 | 541 |
| | NeSVoR | 29.08 | **0.961** | 0.966 | 477 |
| | GSVR (Ours) | **29.74** | 0.952 | **0.971** | **89** |
| CC00629XX19 | SVRTK | 19.35 | 0.729 | 0.734 | 419 |
| | NeSVoR | **26.41** | 0.911 | **0.953** | 480 |
| | GSVR (Ours) | 26.26 | **0.917** | 0.946 | **41** |
| CC00689XX22 | SVRTK | 18.15 | 0.574 | 0.436 | 763 |
| | NeSVoR | 28.79 | **0.940** | 0.957 | 471 |
| | GSVR (Ours) | **28.96** | 0.937 | **0.958** | **98** |
| CC00718XX17 | SVRTK | 17.62 | 0.611 | 0.525 | 218 |
| | NeSVoR | 29.00 | **0.959** | **0.966** | 491 |
| | GSVR (Ours) | **29.01** | 0.948 | 0.965 | **35** |
| CC00735XX18 | SVRTK | 19.01 | 0.710 | 0.617 | 421 |
| | NeSVoR | 28.80 | 0.938 | 0.962 | 481 |
| | GSVR (Ours) | **30.25** | **0.958** | **0.971** | **75** |
| CC00792XX18 | SVRTK | 20.15 | 0.663 | 0.595 | 623 |
| | NeSVoR | **29.24** | **0.958** | **0.955** | 475 |
| | GSVR (Ours) | 28.98 | 0.939 | 0.947 | **95** |
| CC00891XX18 | SVRTK | 19.24 | 0.655 | 0.608 | 806 |
| | NeSVoR | 28.56 | **0.936** | **0.960** | 474 |
| | GSVR (Ours) | **28.60** | 0.931 | 0.959 | **133** |
| CC00979XX23 | SVRTK | 16.43 | 0.541 | 0.337 | 593 |
| | NeSVoR | 25.83 | 0.868 | 0.932 | 480 |
| | GSVR (Ours) | **26.36** | **0.907** | **0.939** | **66** |
| CC01045XX15 | SVRTK | 20.43 | 0.754 | 0.617 | 610 |
| | NeSVoR | 30.42 | 0.954 | 0.963 | 475 |
| | GSVR (Ours) | **31.04** | **0.956** | **0.967** | **64** |
| CC01208XX12 | SVRTK | 18.01 | 0.560 | 0.363 | 738 |
| | NeSVoR | 27.69 | 0.902 | 0.939 | 476 |
| | GSVR (Ours) | **28.38** | **0.916** | **0.947** | **109** |

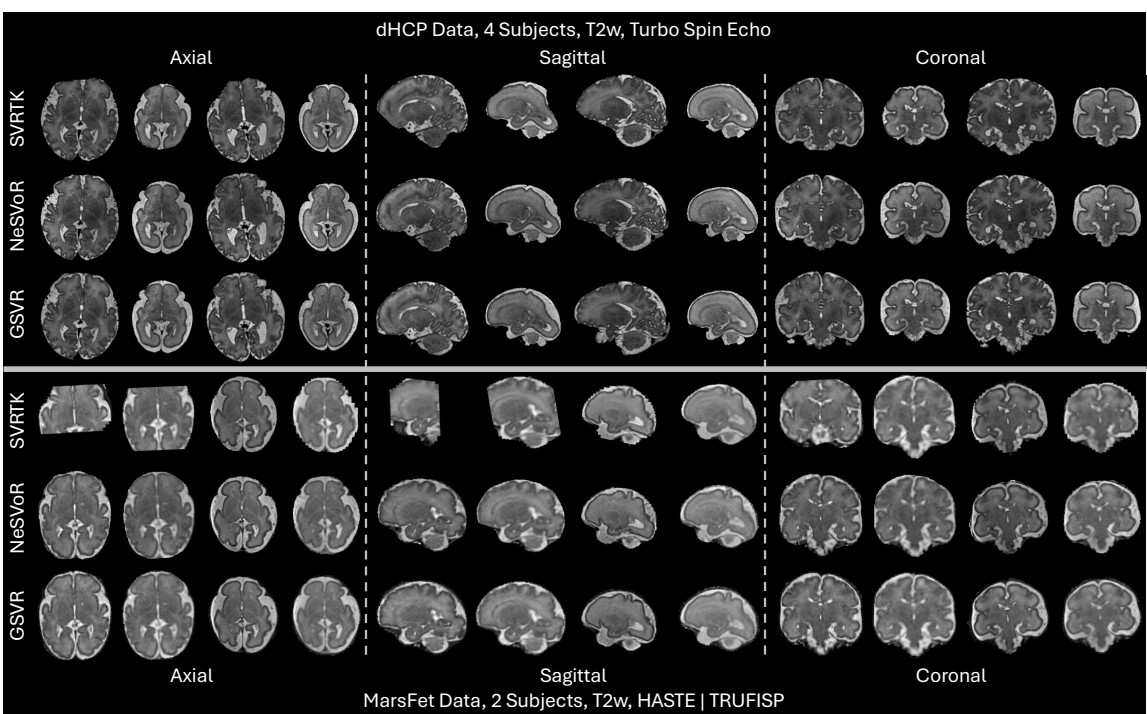

Figure 6: **Qualitative comparison on dHCP and MarsFet cohorts (complete view).** SVRTK exhibits masking artifacts on limited FOVs. The proposed GSVR matches the high reconstruction quality of NeSVoR while achieving speed-ups of up to factor 10.

# Appendix C. Nomenclature

| Symbol | Description |
|--------|-------------|
| *Indices and Dimensions* | |
| $N$ | Total number of Gaussian primitives ($N = 50,000$). |
| $K$ | Number of nearest neighbors for Top-K sparsity ($K = 50$). |
| $i$ | Index for the 2D MRI slice ($i \in \{1, \ldots, M\}$). |
| $j$ | Index for the Gaussian primitive ($j \in \{1, \ldots, N\}$). |
| $\mathbf{u}$ | 2D coordinate in the slice image plane ($\mathbf{u} \in \mathbb{R}^2$). |
| $\mathbf{x}$ | 3D coordinate in the canonical world volume ($\mathbf{x} \in \mathbb{R}^3$). |
| *Gaussian Representation* | |
| $V(\mathbf{x})$ | Continuous volumetric representation of the 3D image volume. |
| $G_j$ | The $j$-th anisotropic 3D Gaussian primitive. |
| $\boldsymbol{\mu}_j$ | Mean position vector of Gaussian $j$ ($\in \mathbb{R}^3$). |
| $\boldsymbol{\Sigma}_j$ | Covariance matrix of Gaussian $j$ ($\in \mathbb{R}^{3\times3}$). |
| $\mathbf{s}_j$ | Scaling vector of Gaussian $j$ (log-space parameters). |
| $c_j$ | Learnable intensity/color parameter of Gaussian $j$. |
| *Forward Model & Physics* | |
| $I_i, \hat{I}_i$ | Observed and reconstructed intensity for slice $i$. |
| $\sigma_i$ | Learnable intensity scaling factor for slice $i$. |
| $\omega_i$ | Aleatoric uncertainty weight for slice $i$ (outlier handling). |
| $\mathcal{T}_i$ | Rigid motion transformation ($\mathbf{u} \to \mathbf{x}$) defined by $\{\mathbf{R}_i, \mathbf{t}_i\}$. |
| $\mathbf{v}_j$ | Centered relative coordinate vector ($\mathcal{T}_i(\mathbf{u}) - \boldsymbol{\mu}_j$). |
| $\phi$ | The 3D Point Spread Function (PSF). |
| $\boldsymbol{\Sigma}_{\text{PSF}}$ | PSF covariance in slice coordinates (thickness/in-plane). |
| $\boldsymbol{\Sigma}_{obs,j}$ | Effective "observed" covariance ($\boldsymbol{\Sigma}_j + \mathbf{R}_i \boldsymbol{\Sigma}_{\text{PSF}} \mathbf{R}_i^T$). |
| *Optimization & Hyperparameters* | |
| $\lambda_{reg}$ | Weight for scale regularization term ($2.5 \times 10^{-3}$). |
| $\mathbf{s}_{target}$ | Target scale hyperparameter (isotropic 1.6mm). |
| $\lambda_{init}$ | Mixing parameter for gradient-adaptive initialization. |
| $\gamma$ | Visualization shrink factor for Gaussian iso-surfaces. |

Table 5: Summary of notation and symbols used in the forward model and optimization.

