# OpenReview forum: "Fast and Explicit: Slice-to-Volume Reconstruction via 3D Gaussian Primitives with Analytic Point Spread Function Modeling"
_MIDL.io/2026/Conference — MIDL 2026 Poster_

### Official Review · Reviewer_ehCo · 2025-12-25

**Confidence:** 5
**Preliminary Rating:** 5
**Final Rating:** 5

**Summary:**

This paper introduces a fast SVR framework for fetal MRI that replaces INRs with explicit 3D Gaussian primitives. By leveraging the mathematical property that the convolution of two Gaussians is another Gaussian, the authors derive a closed-form analytical solution for the Point Spread Function (PSF), eliminating the need for computationally expensive Monte Carlo sampling.

**Strengths:**

1. Achieves substantial acceleration compared to NeSVoR, providing significant benefits to researchers in the community.
2. Identifies the fundamental bottleneck limiting NeSVoR's speed: "reliance on stochastic Monte Carlo sampling to approximate the PSF."
3. The authors suggest the method shows promise for reconstructing Diffusion Weighted Imaging (DWI) data.

**Weaknesses:**

1. The open-source code lacks a comprehensive tutorial; it is hoped that this will be addressed upon paper acceptance.
2. Comparison with NiftyMIC appears to be absent. This method should serve as a baseline for traditional approaches, as studies have indicated that NiftyMIC achieves superior image quality compared to NeSVoR and SVRTK (reference: "Biometry and volumetry in multi-centric fetal brain magnetic resonance imaging: assessing the bias of super-resolution reconstruction").
3. Quantitative comparisons lack objective metrics on real datasets. It is recommended to employ the FetMRQC_SR model to assess the quality of reconstructed images (https://github.com/Medical-Image-Analysis-Laboratory/fetmrqc_sr).

**Detailed Comments:**

N/A

**Justification Of Final Rating:**

This study has received unanimous support from the reviewers; accordingly, I maintain my score and recommend acceptance. Nevertheless, I encourage the authors to conduct more extensive and quantitative image-quality evaluations in future work (e.g., using FetMRQC and/or reader-based assessments).

**Justification Of The Preliminary Rating:**

It is hoped that the code repository will be further refined and ideally distributed as a Docker container to facilitate adoption by community researchers. Future work should include more comprehensive quantitative analyses of reconstruction quality and success rates.

**Questions To Address In The Rebuttal:**

I suggest the authors consider the issues outlined in the weakness section, which could also be addressed in subsequent research.

---

> ### Author Response · Authors · 2026-01-21
>
> We thank the reviewer for the overall positive feedback and are glad the value of the work has been recognized. In the following we want to address the reviewer’s remaining concerns.
>
> - We are currently in the process of restructuring and commenting the code for broad and easy use by the community. We also plan to provide a comprehensive Readme and Docker Container for “plug-and-play”.
> - We agree with the reviewer that in the scope of an extension paper, the study could benefit further from more detailed evaluation including additional (unsupervised) metrics, especially for real-world data,  and additional baselines like NiftyMIC.

---

### Official Review · Reviewer_PyWq · 2026-01-09

**Confidence:** 4
**Preliminary Rating:** 4
**Final Rating:** 5

**Summary:**

The study aims to reconstruct three-dimensional volumetric fetal brain data MRI from motion-corrupted 2D slice acquisitions.
Due to motion and machine related non-homogeneities, they propose efficient and intuitive methods to correct the distorted slice images in three-dimensional space.
The study adapts ImageGS, originally defined on 2d images, to reconstruct 3d volumetric intensity.
The results are compared against an iterative optimization and a recent neural implicit representation model.

[1] Y. Zhang et al., “Image-GS: Content-Adaptive Image Representation via 2D Gaussians,” in SIGGRAPH ’25. doi: 10.1145/3721238.3730596.

**Strengths:**

Intuitive methods are presented for joint processing of point spread function and motion correction, utilizing gaussians as primitives.

The study introduces strong efficiency gains in the number of model parameters and convergence speed compared to recent study NeSVoR, especially important for Fetal data.

Ablation of the number of gaussians, the number nearest neighbours, initialization, point spread function and regularization is studied.

**Weaknesses:**

The results report mean and standard deviation. Adding subject-level results can showcase the stability of the model performance. Furthermore, normalized RMSE metric reported in NeSVoR should also take place in the results for a fair comparison.

In "Towards Real-Time SVR" subsection of the discussion section, authors define "usable" reconstructions for quality control. Can this study extend to criteria of healthy and non-healthy brain differences that the reconstruction enables to distinguish.

Some claims are not substantiated, see detailed comments.

**Detailed Comments:**

While the improvement over INRs wrt applying PSF in the specific is a contribution of the study, the broad impact claim, such as "... any inverse problem involving Gaussian kernels", is not possible to infer from this study. "Analytic Tractability and Broader Impact" section should be explained in further detail regarding how it is inferred from the current study or toned down about the broader impact claims. Fetal brain data is highly distorted that results in both complex and simpler models reaching similar results due to blurred reconstructions. As a result, although the method has potential, its broader impact statement should also consider this fact.

**Justification Of Final Rating:**

The additional experiments on varying levels of motion corruption provide valuable insight and strengthen the paper's conclusions. The proposed method represents a significant and practical advancement in accelerating slice-to-volume reconstruction, is elegantly presented, and is now supported by a more thorough analysis.

**Justification Of The Preliminary Rating:**

While not a foundational study, the suggested adaptation is not straightforward and has high potential impact on further studies in the medical field.  In its current state, the study does not rule out the inherent tendency to select simpler hypotheses due to motion distortion induced limited information. I can increase my rating if the authors can provide a synthetic or real data experiment that shows the performance of the suggested model under varying levels of distortion.

**Questions To Address In The Rebuttal:**

Could a region-specific implicit neural representation be modified to account for PSF and motion correction in a more efficient way than iteratively running for each ROI?

The motion induced limited information in the dataset might be the source of the higher performance compared to a more complex model. Can you provide a performance comparison on varying levels of distortion?

---

> ### Author Response · Authors · 2026-01-21
> **Part 2**
>
> # Questions :
>
> 1.
> - **Question:** Could a region-specific implicit neural representation be modified to account for PSF and motion correction in a more efficient way than iteratively running for each ROI?
> - **Reply:** We interpret the question as if there is a possibility to apply the analytical solution to INRs. The work *AutoInt,* Lindell et al., CVPR 2021 explores analytical integration of INRs which is mandatory for an analytical PSF approach via INRs. AutoInt, however, is currently tailored for 1D line integrals (ray casting) and does not trivially extend to the 3D volumetric convolutions required for our application. On top of that, the approach is highly complex and cumbersome, involving pre-defined strict architectures and reassembling of gradient networks to integration networks inducing additional overhead.
>
> 2.
> - **Question:** The motion induced limited information in the dataset might be the source of the higher performance compared to a more complex model. Can you provide a performance comparison on varying levels of distortion?
> - **Reply:** We thank the reviewer for this question and agree with its relevance. We primarily interpret 'distortion' as inter-slice motion occurring during acquisition. To address this, we included additional experiments on simulated data reducing motion corruption by a factor 1/2 and factor 0, i.e. no motion corruption at all. The results are added as a table to the appendix. And indeed, these experiments confirm the reviewer’s hypothesis. While GSVR still outperforms (no significance) NeSVoR for the *motion corruption factor 1/2* experiment, NeSVoR is slightly, yet significantly, better than GSVR for the artificial setup with no motion corruption. We mainly attribute this difference to the vastly larger number of parameters of NeSVoR compared to GSVR (5 million vs 0.5 million parameters). It is noteworthy, however, that both NeSVoR and GSVR yield very high accuracy with PSNR > 33 dB and SSIM ≥ 0.98. Moreover, we are optimistic that the still relatively new field of Gaussian Splatting / Primitives will achieve further reconstruction improvements in the near future.
> - Finally, we acknowledge that distortion can also refer to intra-slice artifacts (e.g., noise, bias fields, signal dropout). While relevant, thoroughly characterizing the algorithm's robustness to these complex factors is non-trivial and is planned for a dedicated section in a future journal extension.

---

> > ### Comment · Reviewer_PyWq · 2026-01-22
> >
> > Thank you for the detailed clarifications regarding the simulated motion correction levels and the subject-wise results. Your response has largely addressed my concerns about the broader impact of this work. Given its potential significance and relevance to the audience, I am positively inclined to raise my score.
> >
> > To further strengthen an already compelling paper, I suggest one enhancement to the analysis. The comparison could benefit from a more rigorous investigation of the implicit neural representation network (INR) baseline, NeSVoR. Its notable performance degradation with increased noise suggests it might be overfitting to spurious features. Evaluating this baseline with lower-resolution input data or gradually lower number of parameters in exponential scale would be insightful, as it could help distinguish whether the observed performance gaps are attributable to the inherent multi-scale challenges of the problem or to the fundamental methodological advantages of GSVR.

---

> > > ### Author Response · Authors · 2026-01-25
> > >
> > > We sincerely thank the reviewer for the positive assessment and the decision to raise the score. Regarding the suggestion to investigate NeSVoR’s architecture (via parameter reduction or resolution changes), we agree that dissecting the baseline's failure modes is an interesting direction. However, we believe that a detailed ablation study of the baseline’s architecture falls outside the scope of this conference paper, which prioritizes the introduction and validation of the GSVR framework. We plan to include a more granular analysis of baseline behaviors, including potential overfitting, in the future journal extension of this work.

---

> ### Author Response · Authors · 2026-01-21
> **Part 1**
>
> We are happy that the reviewer acknowledges the potential and merit of the proposed work. We also want to thank the reviewer for the still critical feedback which helped us to further improve the paper. In the following we are going to address the feedback point by point.
>
> - **Inter-subject performance:** The mean and standard deviation (std) are reported over all subjects. Hence, std can be viewed as an accurate statistical metric for inter-subject variance, hence performance stability. However, for a fine-grained performance evaluation, we have now added the individual subject results as Table 4 to the appendix of the revised manuscript.
> - **NRMSE as additional metric:** We agree that NRMSE is a valid metric. However, given that NRMSE and PSNR are functionally equivalent for ranking model performance (as both rely on MSE), we opted to report only PSNR. Still, for more convenient evaluation for the reader, we have now added NRMSE to the results of table 1 in the revised man.
> - **Towards Real-Time SVR:** This is probably a misunderstanding. The preliminary results are not meant for classification. They are rather meant as indicators for the radiologist to assess during the acquisition if the reconstruction quality is sufficient or if additional slice acquisitions are necessary for a good reconstruction.
>
> # Detailed Comments:
>
> 1.
> - **Comment by Reviewer:** "While the improvement over INRs wrt applying PSF in the specific is a contribution of the study, the broad impact claim, such as '... any inverse problem involving Gaussian kernels', is not possible to infer from this study. "
> - **Reply:** We toned down the broad impact claim and explicitly stated that these claims need to be addressed further in future work.
>
> 2.
> - **Comment by Reviewer::** “Fetal brain data is highly distorted that results in both complex and simpler models reaching similar results due to blurred reconstructions.”
> - **Reply:** We interpret the reviewer’s comment regarding “highly distorted” as referring to the magnitude of subject motion. We agree that distortion factors like motion corruption can indeed obscure the model's super-resolution performance. To address this, we included additional experiments on simulated data reducing motion corruption by a factor 1/2 and factor 0, i.e. no motion corruption at all. The results are added as a Table 3 to the appendix.
> Indeed, these experiments confirm the reviewer’s hypothesis. While GSVR still outperforms (no significance) NeSVoR for the *motion corruption factor 1/2* experiment, GSVR is slightly but significantly worse than NeSVoR for the experiment with no motion corruption. We mainly attribute this difference to the vastly larger number of parameters of NeSVoR compared to GSVR (5 million vs 0.5 million parameters). Finally, it is noteworthy that both NeSVoR and GSVR yield very high accuracy with PSNR > 33 dB and SSIM ≥ 0.98.

---

### Official Review · Reviewer_mo6m · 2026-01-10

**Confidence:** 3
**Preliminary Rating:** 5
**Final Rating:** 5

**Summary:**

This paper describes an adaptation of 3D Gaussian splatting to slice to volume problems in medical imaging, with a focus on motion-corrupted fetal MRI reconstruction. Compared to previous state of the art approaches that require Monte Carlo methods to perform the intractable integration step, the Gaussian splatting approach has a closed form solution. This leads to results that are similar to the current state of the art but consistently faster.

**Strengths:**

The results of this paper are a significant achievement in accelerating the SVR problem, a topic of considerable importance within medical imaging. The method is simple and elegant. The experimental results are compelling, and further ablations offer further insight into the method. The paper is very clearly written and well presented.

**Weaknesses:**

There are no major weakness. In some sense, the novelty is limited because it is a fairly straightforward application of 3DGS from the graphics community to this task, with a minor difference to account for the different 2D acquisition geometries (projective vs volumetric).

**Detailed Comments:**

- It would be helpful to differentiate between the modelled intensity of the plane and the observed intensity. Currently $\hat{I}$ is only introduced in equation 7 (and not actually defined)

**Justification Of Final Rating:**

I think my initial rating still stands. I believe this is an important result for the community, since it accelerates the SVR problem by 5-10x. Furthermore the paper is very well written and clearly presented, the method is elegant and the results are compelling

**Justification Of The Preliminary Rating:**

I believe this is an important result for the community, since it accelerates the SVR problem by 5-10x. Furthermore the paper is very well written and clearly presented, the method is elegant and the results are compelling

**Questions To Address In The Rebuttal:**

No questions

---

> ### Author Response · Authors · 2026-01-21
>
> We thank the reviewer for the generally positive feedback and appreciation of our work.
> Regarding novelty, we view the contribution to be two-fold.
>
> * First, the reformulation of the 2D gaussian splatting approach to a 3D super-resolution projection-free framework with explicit regularization.
>  * Second, the analytical PSF for which, to the best of our knowledge, we are the first to exploit the closed-form property of gaussians under convolution in this setting.
>
> Finally, we would also like to thank the reviewer for the valid feedback on the inconsistencies regarding the definition of both $I$ and $\\hat{I}$.
> We have now modified the methods section, including equations, to consistently treat $I(.)$  as observed (ground-truth) intensity and $\hat{I}(.)$ as estimated/modeled intensity.

---

### Author Rebuttal · Authors · 2026-01-21

**Rebuttal:**

We thank all reviewers for their valuable time and constructive feedback. In the following, we try to adequately address all remaining concerns. Additionally, we provide to the reviewers the revised markup manuscript. All changes are colored in red.

**Supporting Material:**

/attachment/a2d8077feba17c8aaf1b95f3d5088afe92d8bf27.pdf

---

### Meta-Review · Area_Chair_2DUe · 2026-02-09

**Recommendation:** Accept (Oral)
**Confidence:** 5

**Metareview:**

This paper presents a fast slice-to-volume reconstruction (SVR) framework for fetal MRI based on adapting 3D Gaussian splatting with an analytical, closed-form treatment of the point spread function. Reviewers consistently viewed the approach as elegant, technically sound, and highly relevant, with a clear practical benefit in substantially accelerating SVR while maintaining competitive reconstruction quality. The simplicity of the formulation, strong efficiency gains over INR-based methods, and clear experimental validation were repeatedly highlighted as major strengths. The paper was also praised for its clarity, presentation quality, and insightful ablation studies.

The concerns raised by reviewers were largely constructive and focused on scope and completeness rather than fundamental flaws. These included requests for additional reporting of subject-level performance, inclusion of complementary metrics such as NRMSE, clarification and moderation of broad impact claims, evaluation under varying motion or distortion levels, and suggestions for additional baselines or deeper analysis of competing methods. Minor issues related to notation clarity and code usability were also noted.

The authors responded comprehensively to these points. They added subject-wise results, included additional metrics, performed new experiments across varying motion corruption levels, clarified notation and modeling assumptions, and appropriately toned down broad impact claims. Where reviewers suggested more extensive analyses or additional baselines, the authors provided clear justifications for deferring these to future journal extensions, which reviewers generally found reasonable. These revisions strengthened the manuscript and addressed the key concerns raised during review.

Overall, the reviewers agreed that this work represents a significant and practical advance in fast SVR for fetal MRI. The authors responded constructively and comprehensively to reviewer feedback, strengthening the manuscript through additional experiments, clearer claims, and improved reporting. While some extensions were deferred to future work, this was considered appropriate for the venue, and all reviewers ultimately supported acceptance.

---

### Decision · Program_Chairs · 2026-02-13

Accept (Poster)